# Effects of Self-Rated Health Status on Residents’ Social-Benefit Perceptions of Urban Green Space

**DOI:** 10.3390/ijerph191610134

**Published:** 2022-08-16

**Authors:** Yuhong Tian, Fenghua Liu, Chi Yung Jim, Tiantian Wang, Jingya Luan, Mengxuan Yan

**Affiliations:** 1State Key Laboratory of Earth Surface Processes and Resource Ecology, School of Natural Resources, Faculty of Geographical Science, Beijing Normal University, Beijing 100875, China; 2Department of Social Sciences, Education University of Hong Kong, Tai Po, Hong Kong, China; 3School of Environment and Nature Resources, Renmin University of China, Beijing 100872, China

**Keywords:** urban green space (UGS), social benefit, self-rated health status, physical health benefit, resident perception, living environment

## Abstract

Urban green spaces (UGS) provide many social benefits and improves residents’ wellbeing. Studying residents’ perceptions of UGS’s social benefits and driving factors could promote public health and environmental justice. A questionnaire survey of 432 Beijing residents and statistical tests assessed the impacts of residents’ living environments and self-rated health status on UGS perceptions. The results showed: (1) perceptions of UGS’ physical health benefits were subdued, with an inclination towards other social benefits. Respondents more highly perceived accelerating patient recovery and reducing morbidity and mortality rates. Perceptions of bearing larger-head babies with higher weight were relatively low. For other social benefits, perceptions of improving the environment and life quality were higher, but reducing anger outbursts and resolving conflicts were lower. (2) Childhood living environments did not affect perceptions of social benefits, but current living environments did. Suburb residents understood reducing pain-relief medication demands and bearing larger-head babies better than city residents. City residents understood UGS’ investments considerable and sustained returns better than village residents. City residents agreed with accelerating patient recovery higher than village ones. (3) Respondents with “poor” self-rated health status had better perceptions of other social benefits. Those with “excellent” ratings did not fully understand UGS’ physical health benefits. “Poor” ratings understood improving a city’s image and making cities livable and sustainable better than “good” or “fair” ratings. “Excellent” ratings had less understanding of larger-head babies than “good” or “fair” ratings. The study could enhance appreciation of UGS’ social benefits to facilitate planning and management to meet residents’ expectations.

## 1. Introduction and Literature Review

Green spaces are the surfaces and spaces where vegetation grows, occupies, and covers land [1]. Urban green spaces (UGS) usually include four categories with reference to land use and accessibility: public green spaces (usually referring to urban parks), semi-public green spaces (such as open spaces within the grounds of hospitals, government, institutional, or private sector facilities, which fewer people can use), private green spaces (referring to residential garden maintained by residents), and street green spaces [2].

The provenance of green sites in a city includes a spectrum of naturalness, from entirely natural enclaves inherited from pre-urbanization ecosystems to modified and created nature in a continuum of biotopes [3]. In terms of management, they can range from meticulously manicured to minimal or no human input. The former is often considered formal UGS and the latter informal. In general terms, they include parks, gardens, woodlands, nature areas, and other green sites. UGS is defined broadly in this study as the publicly-owned open space in urban areas with a good vegetation cover and sometimes embedded water bodies, providing access to the public and natural ingredients in different forms [4]. Most UGS used frequently by residents are located at or near urban precincts with relatively high population density. Thus, most people’s exposure, experience, and perception of UGS are associated with these popular UGS sites. Other green spaces on the urban fringe can be accessed by urban residents willing to travel to reach them. Such green space, situated in areas with more countryside cover than built-up areas, differs from urban green space, usually surrounded by the dominating gray infrastructure [5].

As essential components of urban ecosystems, UGS are often the few or only places where people can experience nature [6]. Such green sites are often known as the “lung of the city” [7]. They provide residents with many benefits and services and contribute significantly to the city’s life support system. Living under the shadow of urbanization, pollution, traffic congestion, and poor housing quality has threatened residents’ health and wellbeing. UGS provide essential relief from the often-dominating gray infrastructure in the form of many social benefits, including physical health. They play vital roles in improving residents’ wellbeing and sustainable urban development.

Health is the crucial means to achieve development, and the ultimate development goal. It aims to ensure a healthy life and promote the wellbeing of citizens of all ages [8]. Greenery embedded in urban areas can impart a wide range of health benefits. At the local scale, UGS visits can enhance residents’ health by increasing physical activities in the salubrious open-air ambiance [9,10], relieving psychological stress, and expediting fatigue recovery [11]. More people of diverse socio-demographic profiles visiting UGS can promote social contacts and strengthen social connection, cohesion, and harmony [12,13]. Vegetation cover and natural elements can alleviate urban environmental risks, moderate microclimate, and mitigate urban heat island and climate change impacts [14,15,16].

As a specific land use, hospitals accommodate mainly medical and other staff, patients, and visitors. Patients suffering from health issues are often physically and psychologically stressed and vulnerable. It is important to reduce stress to accelerate patients’ postoperative recovery and improve their quality of life during hospitalization [17]. Green spaces can bestow multiple health and healing functions: (1) providing distraction through vision and a relaxed ambiance; (2) serving as an air purifier and humidifier in the indoor environment by absorbing volatile organic compounds and other air pollutants; (3) actively improving the physiological reactions of patients, such as heart rate, blood pressure, respiratory rate, and muscle tension, and reducing the consumption of analgesics and nursing care; (4) indirectly relieving stress by promoting social support (such as providing social places to enhance self-confidence and prevent loneliness, etc.), which has a significant impact on patient health enhancement.

The presence of natural elements in a hospital setting, outdoor and indoor, has therapeutic effects on patients. Ulrich’s pioneering study found that a natural green view through a window could provide the restorative capability to expedite patient recovery from surgery and demand less potent analgesic medication [18]. Khan et al. studied 270 patients in the medical center in Peshawar, Pakistan. They found that the ward with foliage plants and flowers could shorten the postoperative hospitalization time, reduce analgesic intake, lessen pain intensity, reduce stress fatigue, and improve vital signs (blood pressure, heart rate, respiratory rate, and body temperature) [19]. Patients had more positive emotions and feelings during their hospital stay. Yar & Kazemi (2020) assessed 54 hospitalized children in Mashhad, Iran [17]. They found that patients with ready access to green spaces were significantly healthier. Their vital signs, including systolic and diastolic blood pressure, pulse, and respiratory rate, were improved by green space visits. The neuropsychological indices were enhanced by green landscape, including the emotional index (fear, happiness, and relaxation), cognitive index (compatibility and attention), and non-symptoms index (anxiety and depression). Vujcic et al. evaluated 30 psychiatric patients in a Serbian psychiatric hospital. They found that gardens can significantly reduce anxiety, depression, and stress of patients with severe mental illnesses [20].

The health benefits of exposure to greenery can be bestowed even before birth. Pregnancy outcomes directly affect infant health, with notable influences on morbidity and mortality rate in early life. The effects on health and development can linger throughout an individual’s life. For instance, it may increase the risk of ischemic heart disease and chronic diseases in adulthood (such as hypertension, obesity, diabetes, and cardiovascular disease) [21]. Pregnant women visiting UGS frequently can improve their health status, bringing a healthy fetal environment [22,23] and better pregnancy outcomes. Pregnancy outcomes can usually be measured by full-term birth, baby weight, head circumference, and gestational age [23,24,25]. Dadvand et al. examined 2393 singleton live births in Spain in different climates and vegetation patterns [24]. They found that pregnant women’s high level of residential surrounding green space was related to higher birth weight and head circumference. Hystad et al. investigated 64,705 newborns in Vancouver, British Columbia, Canada. They found that the increase in green spaces in residential areas was associated with higher birth weight [26]. Maternal greenspace exposure also reduced the likelihood of a shorter gestational age and very preterm (<30 weeks) and moderately preterm (30–36 weeks) birth. Zhan et al. reviewed 36 studies involving 11,983,089 participants and found that exposure to green spaces increased baby weight and head circumference [21]. Increasing the urban forest cover could reduce preterm birth and raise birth weight [27].

In addition, UGS can also reduce the mortality of residents. Diseases related to limited exposure to green space include respiratory diseases, cardiovascular diseases, prostate cancer, mental diseases, obesity, hypertension, etc. [28,29,30,31,32]. The impacts of UGS on these diseases vary among residents of different gender, regions [33], and socioeconomic levels [34]. Higher tree cover in neighborhoods can improve residents’ general health [35]. Cities with relatively low green cover experienced fewer preventable premature deaths [36]. Conversely, increasing the tree cover of a city could significantly reduce the annual premature mortality burden [37]. UGS can also provide many other important social benefits. They include: (1) improving people’s perception of quality of life [38]; (2) providing meeting places for group activities, such as conversation, group exercise, and picnics, thereby encouraging residents to conduct social exchanges, promote neighborhood relations, and improve community cohesion [39,40]; (3) reducing personal aggression and violent behaviors, and providing sites and situations that are amenable to social surveillance to suppress violent and property crimes [41,42,43]; (4) symbolizing history, culture, beliefs, and collective memory through the medium of the community’s heritage trees [44,45]; (5) creating and strengthening the positive image and local characteristics of the whole city, providing more attractive and livable places, establishing a green city brand to attract talent, tourism, and investment, and bringing sustainable benefits in environmental, social, and cultural realms [46,47].

Perception is the subjective reflection of the subject to the objective environment. It also indicates the object’s meaning to the subject and the value demand of the subject [48]. Research on residents’ perceptions can understand UGS’ values and problems from users’ perspectives. The findings can inform the optimization strategies to improve UGS management. It can fully help realize UGS’ social benefits and improve residents’ enjoyment [49]. Many studies have investigated residents’ perceptions of UGS’ benefits. For example, Chen et al. divided perceptions of green spaces into eight dimensions (serene, nature, rich in species, space, prospect, refuge, social, and culture). They found that residents in Hefei, China, had the highest perception of the social dimension but the lowest of culture and biodiversity [50]. Ko and Son studied the residents’ perceptions of UGS in Gwacheon, Republic of Korea. They found that residents well perceived the cultural ecosystem services of UGS in their daily environment, such as recreation, cultural heritage, aesthetic, educational, spiritual-religious, social relations, and health [51]. Riechers et al. divided the cultural ecosystem services provided by UGS into ten categories (education, nature awareness, aesthetics, cultural heritage, sense of place, religious-spiritual, recreation, inspiration, cultural diversity, and social relations) [52]. Their survey in Berlin, Germany, found that residents accorded the highest importance to aesthetic value, followed by nature awareness and religious-spiritual. Cultural diversity was rated as the least important.

The driving factors of UGS’ benefit perceptions include society, economy, psychology (such as age, gender, education level, social-economic status, ethnic origin, or emotional ties), culture, and climate zone [53]. Many studies focused on the impacts of residents’ current socioeconomic characteristics on perceptions. For example, Wright Wendel et al. conducted a study in Santa Cruz, the largest city in Bolivia [54]. They found that younger residents used green spaces more to entertain and meet others, whereas middle-aged and elderly people preferred to use green spaces to escape daily life pressure and play with children. Abass et al. surveyed the residents of the Ghanaian city of Greater Kumasi [55]. They found that residents’ perceptions of UGS’ benefits were primarily affected by formal education. Fernandes et al. studied the residents’ perception of street trees on the seafront of Porto, Portugal [56]. They found that more educated residents perceived the importance of street trees and had higher perceptions of tree benefits than damages caused by trees. The elderly were more likely to perceive trees as dangerous.

Among the socioeconomic factors, health status and living environments are considered important. Self-related health status can comprehensively evaluate an individual’s health status from psychology, physiology, and society. The health evaluation index commonly uses residents’ self-related health status based on a five-level Likert scale. For example, Romagosa found that tourists who thought they were healthy could better perceive the health benefits of the park in the Barcelona metropolitan area [57]. Bai et al. found that residents’ body mass index (BMI) was related to the perception of UGS’ benefits in the Kansas City metropolitan area [58]. However, few studies paid attention to the UGS system. Furthermore, it has remained unclear how residents’ health status may affect their perceptions of specific UGS’ social benefits.

Childhood is the crucial stage in life, and is the period when a person’s personality and psychological, physiological, and social foundations are formed. The living environment in childhood is important to their formative physical and mental development, with repercussions on happiness and health [59] and long-term effects on their attitude and behavior in adulthood [60,61]. For example, Putra et al. studied 1874 children in Australia and found that children living near high-quality green spaces had a lower hospital admission rate [62]. Acolin et al. assessed 95 children living in Seattle, WA, and found that children living near parks with playgrounds had better mental health [63]. Bezold et al. evaluated over 11,000 teenagers in the United States. They found that residents living in areas with a high green cover in childhood had better mental health. This benefit was especially evident in urban areas by reducing the incidence of depression in adolescence [64]. Browning et al. studied 297 students in American public universities. They found that the emotional intelligence of young people who grew up in relatively low-income areas was positively correlated with the cumulative community greening rate around their childhood homes [65].

The current living environments also affect residents’ perceptions of UGS’ benefits. For example, Romagosa found that residents living near parks can better perceive their health-promoting benefits [57]. Riechers et al. investigated Berlin, Germany, and found that residents living in suburban areas were more inclined to perceive UGS’ benefits related to recreation [52]. Young residents in urban core areas better perceived the UGS’ benefits related to social relations. However, these studies lacked a systematic evaluation of how the living environments at different life stages affected perceptions of specific UGS’ social benefits such as physical health.

UGS can play an important role under the “Healthy China” national strategy [66]. We chose Beijing, the pilot city of the “Healthy China” program, as the case study. Residents’ perceptions of UGS’ social benefits were surveyed. The impacts of residents’ self-rated health status and childhood and adulthood living environments on their perceptions were evaluated. The study aimed to expand the theoretical and practical basis to understand residents’ perceptions of UGS’ social benefits and enhance the planning, development, and management of UGS to meet residents’ needs and expectations. It is hoped that the findings could modify policies using UGS to improve public health, environmental justice, and sustainable and healthy development of cities.

## 2. Study Area and Methods

### 2.1. Study Area

Beijing is China’s capital and political, economic, transportation, and cultural center. The large metropolis has experienced a rapid urban expansion in the past three decades, with the fastest urbanizing rate in China [67]. With a total area of 16,410,000 ha, Beijing includes 16 county-level administrative divisions, and its center is located at 39°54′ N and 116°25′ E. It has northern China’s typical warm and temperate semi-humid continental monsoon climate [68]. Summer is hot and rainy and winter cold and dry, interspersed by a short transitional spring and autumn.

By late 2018, Beijing’s population of permanent residents reached 21.54 million. The total area of UGS was 85,286.37 ha, with a greening rate of 46.17% and a green area supply of 42.15 m^2^/person. Urban parks contribute the largest share of 32,618.50 ha, at 16.30 m^2^/person. The UGS coverage rate within a 500 m radius service zone (residential land area covered by UGS service zone/total residential land area × 100%) attained a high level of 80% [69]. This means that 80% of Beijing residents have urban green space lying within 500 m of their homes. The total UGS area in the main built-up areas (located within the Fifth Ring Road) was 19,569 hectares, with an average green patch size of 16.20 ha [70] (Figure 1). Recent expenditures on greening have further increased the number and types of green spaces, such as city-center small gardens, boulevards, theme parks, and large rural forest parks in the city’s periphery [71]. The preservation of 39,408 large ancient trees in Beijing indicates its long history of association with urban greening [72].

### 2.2. Data Collection

A questionnaire survey is an important data collection method. Based on the research content and previous research results, a questionnaire was prepared and filled by the respondents to collect people’s attitudes and opinions on a specific issue. Each questionnaire contained 27 questions assigned to two parts. The first part was the respondents’ perception of UGS’ social benefits. It included ten specific benefits of improving physical health and ten other social benefits. We divided health benefits into three aspects: hospital patients, pregnancy outcomes, and morbidity and mortality rate (Table 1). Perception includes understanding and agreement. A Likert scale measured the responses to the statements (from “1”: don’t know/don’t agree to “5”: know excellently/agree very strongly). The questionnaire’s second part gathered the respondents’ socioeconomic characteristics, childhood and current living environments, and self-rated health status.

In the summer of 2018, we conducted a questionnaire survey. We established a stratified and purposeful sampling scheme to obtain a representative sample of Beijing residents. According to their monthly household income (2017), potential respondents were divided into six categories, namely RMB <6000 (about USD 907.47 in 2017), RMB 6000–14,000 (about USD 2117), RMB 14,000–24,000 (about USD 3628.87), RMB >24,000, unemployment and retirement. The percentage of Beijing residents in the income categories (2017) was obtained from the website (http://www.sohu.com/a/168513231_99914104) (accessed on 18 March 2018). The data were obtained from the field investigations of the above 5000 samples for each occupation. In each income category, representative occupations were identified, and the target number of respondents for each occupation was estimated based on census data. Work units with potential respondents were randomly determined and selected according to the respondent distribution plan. The questionnaires were distributed to these units, such as hospitals, offices, factories, parks, etc. A total of 500 completed questionnaires were collected from the work units. After eliminating the invalid questionnaires, 432 questionnaires were used for subsequent statistical analysis.

Before data analysis, all the returned questionnaires were carefully screened to eliminate those judged invalid, including incomplete or missing entries, responses lying outside the expected range, and contradictory or incorrect answers. The response rate was 86.4%, regarded as an acceptable or respectable outcome in social science research. Chi-square tests were employed to assess the differences in main socioeconomic characteristics between the survey respondents and Beijing residents (Table 2). The results indicated that gender and marriage were similar. In the questionnaire survey, only residents aged above 18 were investigated, therefore their age and education levels were different from the city’s residents.

### 2.3. Data Analysis

Descriptive and inferential statistical analyses were performed using Microsoft Excel (Microsoft, Redmond, WA, USA) and IBM SPSS 20.0 software (IBM Corp., Armonk, NY, USA). For the current perception situation, the percentage of respondents in each perception level in the total number of respondents was calculated for each question. The average perception score was calculated by multiplying the percentage by the corresponding perception measurement value (1–5). In each category, a bar graph was drawn according to the average perception score. The average of ten terms was shown on the horizontal axis, and columns exceeding and below the average were arranged above and below the average line, respectively (Figure 1).

Preliminary analysis showed that the sample data were not normally distributed, so non-parametric statistical tests had to be used. Although non-parametric tests are generally not as powerful as parametric tests, they can yield reliable results with sufficiently large sample sizes. The Kruskal–Wallis and Mann–Whitney tests were used to compare residents’ perceptions of benefits in different living environments and self-rated health conditions. Such comparisons can allow an objective judgment of the differential effects of self-rated health status on residents’ social-benefit perceptions of urban green space, which is the gist of this study. A statistically significant difference was set at *p* < 0.05 and marked with *. A strong statistically significant difference was set at *p* < 0.01 and marked with **. The Kruskal–Wallis test was applied first to compare pairs of variables. If the results showed significant differences, the Mann–Whitney test was then applied for intra-group comparison involving three or more variables.

## 3. Results

### 3.1. Perceptions of UGS’ Social Benefits

Table 3 shows respondents’ perceptions of various social benefits. About 33.43% of the respondents had a “fair” understanding of physical health benefits, followed by those with “limited” understanding (29.44%). Only 7.22% and 8.13% of respondents had “excellent” and “don’t know” understanding of physical health benefits, respectively. Respondents’ understanding of B1-1 to B1-3, B1-8, and B1-9 were higher than the average value (Mean = 2.91). Most respondents indicated the strongest agreement with B1-1 (MD = 4, RI = 0.79) and B1-8 (MD = 4, RI = 0.74). They had the lowest agreement degree of B1-6 (RI = 0.59). Their understanding of B1-1 (Mean = 3.54, MD = 4, RI = 0.71) scored the highest value, and of B1-5 (Mean = 2.64, MD = 2, RI = 0.53) and B1-6 (Mean = 2.40, MD = 2, RI = 0.48) the lowest values. The respondents with “moderate” agreement contributed the highest proportion (35.25%), followed by “strong” agreement (27.52%). The respondents with “don’t agree” had the lowest (4.05%). Respondents’ agreement with B1-1, B1-8, and B1-9 were higher than the average (Mean = 3.33). Their agreement of B1-1 (Mean = 3.92) was the highest, and of B1-5 (Mean = 3.09) and B1-6 (Mean = 2.94) was the lowest. The rank of respondents’ understanding of physical health benefits was similar to their agreement. Respondents’ scores for understanding degrees of B1-7 (SD = 1.17), B1-5 (SD = 1.09), and B1-10 (SD = 1.06) were scattered. Their scores of understanding of B1-1 (SD = 0.91), B1-2 (SD = 0.96), and B1-3 (SD = 0.98) were relatively concentrated. The respondents’ scores of agreement degree for B1-10 (SD = 1.11) and B1-4 (SD = 1.07) were scattered. Their scores of B1-1 (SD = 0.86) and B1-8 (SD = 0.97) were relatively concentrated.

About 40.25% of respondents had a “good” understanding of other social benefits, followed by those with a “fair” understanding (26.11%). The respondents with a “don’t know” understanding accounted for the lowest proportion (0.81%). Respondents’ understanding of B2-1, B2-2, and B2-7 to B2-9 were higher than the average value (Mean = 3.76). The respondents’ understanding of B2-1 (Mean = 4.06, MD = 4, RI = 0.81), B2-2 (Mean = 4.05, MD = 4, RI = 0.81), and B2-9 (Mean = 4.03, MD = 4, RI = 0.81) were relatively high compared to other social benefits. Their understanding of B2-5 was low (Mean = 3.23, MD = 3, RI = 0.65). The respondents expressing “strong” agreement took the largest share of 38.29%, followed by “very strong” agreement (35.58%). The respondents with “don’t agree” were the lowest at merely 0.95%. Respondents’ agreement with B2-1, B2-2, and B2-7 to B2-9 were higher than the average (Mean = 4.02). Their agreement of B2-1 (Mean = 4.29) and B2-2 (Mean = 4.29) took the highest proportions, and their agreement of B2-4 (Mean = 3.79) and B2-5 (Mean = 3.53) were the lowest. They had the highest agreement for B2-1 (RI = 0.86), B2-2 (RI = 0.86), and the lowest agreement for B2-5(RI = 0.71). The ranks of respondents’ understanding and agreement of other social benefits were similar. Respondents’ scores for B2-5 (SD = 0.99), B2-4 (SD = 0.98), and B2-10 (SD = 0.97) were relatively scattered, and the scores for B2-1 (SD = 0.82), B2-2 (SD = 0.83), B2-9 (SD = 0.84) were relatively concentrated; the respondents’ scores of agreement degrees for B2-5 (SD = 1.02) and B2-4 (SD = 1.00) were scattered, and for B2-1 (SD = 0.73) and B2-2 (SD = 0.76) were relatively concentrated. Overall, the respondents’ understanding of B1 (MD = 3, RI = 0.58) was lower than B2 (MD = 4, RI = 0.75). Most respondents had higher agreement of B2 (MD = 4, RI = 0.80) than B1 (MD = 3, RI = 0.67).

### 3.2. Impacts of Living Environments on Perceptions of UGS’ Social Benefits

Table 4 shows the impacts of childhood living environments on residents’ understanding of and agreement with UGS’ social benefits, respectively. The differences in understanding and agreement degree of all UGS’ social benefits among respondents with different childhood living environments were statistically insignificant.

Table 5 shows the influence of current adulthood living environments on residents’ understanding of and agreement with UGS’ social benefits, respectively. For physical health benefits, city respondents had a significantly lower understanding of B1-3 and B1-6 than suburb ones (*p* = 0.005 ** and *p* = 0.014 *, respectively). City and suburb respondents had a significantly higher agreement of B1-1 than village ones (*p* = 0.025 * and *p* = 0.013 *, respectively). For perceptions of other social benefits, city respondents had a significantly higher understanding of B2-10 than village ones (*p* = 0.013 *). The differences between respondent groups were statistically insignificant for agreement degree by current living environments.

### 3.3. Impacts of Self-Rated Health on Perceptions of UGS’ Social Benefits

Table 6 shows how respondents’ self-rated health status impacts residents’ understanding of and agreement with of UGS’ social benefits, respectively. For physical health benefits, respondents with “excellent” ratings had a significantly lower understanding of B1-6 than those with “good” and “fair” ratings (*p* = 0.033 * and *p* = 0.010 *, respectively). The differences in respondents’ agreement degree were statistically not significant.

For other social benefits, respondents with “excellent” ratings had a significantly higher understanding of B2-8 and B2-9 than those with “good” ratings (*p* = 0.015 * and *p* = 0.005 **, respectively). Respondents with “bad” ratings had a significantly higher understanding of B2-8 and B2-9 than those with “good” ratings (*p* = 0.020 * and *p* = 0.021 *, respectively) and “fair” ratings (*p* = 0.017 * and *p* = 0.013 *, respectively). In addition, respondents with “excellent” ratings had a significantly higher agreement of B2-8 than those with “good” or “fair” ratings (*p* = 0.004 ** and *p* = 0.027 *, respectively).

## 4. Discussion

### 4.1. Residents Were More Inclined to Perceive Other UGS’ Social Benefits

Many studies have pointed out that green space can provide positive health effects, such as reducing the risk of infectious diseases (dengue fever, etc.), chronic diseases, and maternal and infant diseases, as well as alleviating stress levels and mental diseases (anxiety and depression, etc.) [74]. However, our results found that the residents’ perceptions of UGS’ physical health benefits were not high. Among them, the perceptions of accelerating patient recovery and reducing morbidity and mortality rates were relatively high. Understanding such health benefits has permeated different population segments to become the community’s general knowledge. This type of research on the relatively primary health benefits of urban greening was initiated many decades ago. Research findings have been widely translated into public policies and urban designs [75,76,77]. Many people are well aware of the health dividends of visiting UGS.

In contrast, the perceptions of bearing babies with larger heads and higher birth weight were relatively low. This observation may be related to residents’ changing concepts, understanding, and fertility experience. Due to the high cost of living in big cities like Beijing, there was a growing aversion towards marriage and childbirth and an increasing trend of late marriage and late childbirth [78,79].

These fundamental changes in societal attitudes could reduce exposure and dampen the need to acquire relevant knowledge about pregnant women and pregnancy outcomes. Such issues would be considered irrelevant, lying well beyond the routines or remits of many people’s lives, especially the younger residents who do not wish to have children.

Besides, specific research concerning the prenatal benefits of urban greening has achieved notable advances only recently, often enlisting big data analysis [80,81]. It will take time for the investigations to expand, deepen, and elaborate, and the findings to transfer and translate to the citizen-science realm. The relationship between UGS and pregnancy outcomes could be further explored regarding climatic regions, ethnic differences, city size, urban development, and population densities, and the amount, distribution, and nature contents of UGS. Such knowledge could benefit the portion of residents who want to have babies. It could inform their choice of living location and environment, even before childbirth, to influence the quality of the pregnancy outcome. The research findings could also inform urban design and planning to optimize the quality, quantity, and spatial pattern of UGS provision. Unfortunately, indoor space poverty is often associated with outdoor UGS shortage. Therefore, specific neighborhoods with socioeconomic poverty should receive special attention [82]. Measures such as planning requirements to improve the quality of the built environment and property tax reform may address the environmental injustice of UGS deficiency or deprivation [24,83,84]. City governments can also integrate health-related elements into UGS, such as footpath systems to encourage walking and jogging, health knowledge promotion areas, fitness patches, and collective exercise sites to improve UGS’ attractiveness and patronage [85]. It is worthwhile to nurture a competitive and recognized green-city brand by linking UGS with promoting the municipal goals of improving public health and attracting tourists, talents, businesses, and investments [46,47,86].

For other UGS social benefits, residents had higher perceptions of improving the quality of environment and life but lower perceptions of reducing anger outbursts and resolving conflicts. UGS promotes social communications and interactions and provides venues for informal meetings and group activities. UGS can encourage social gatherings and surveillance to reduce violent and property crimes [43,87]. Human contact with greenery can depress the likelihood of personal aggression and violence to bring harmony within and beyond family units. These subtle personal and community benefits can usher the welcomed collateral social cohesion, stability, and security. In conjunction with organized outdoor activities, publicity and public education endeavors could disseminate the relevant knowledge and raise the perception of the broad spectrum of UGS’ benefits. A more knowledgeable citizenry could expand the beneficiaries’ base of UGS with far-reaching repercussions on public health and happiness [88].

### 4.2. Residents’ Current Living Environments Affected Perceptions of UGS’ Social Benefits

Our results showed that childhood living environments did not significantly affect perceptions of UGS’ social benefits. Childhood exposure to more proximal and better natural ingredients did not influence adulthood perceptions of UGS’ social benefits. On the one hand, even if childhood contact with UGS was insufficient due to the different distribution and usage modes of UGS, adults may actively contact and use UGS and have environmental protection behaviors [89]. They reduce the perceived difference in the process of frequent contact. On the other hand, people’s health might have benefited from childhood village and suburb living, yet they did not associate current UGS’ benefits with health. This apparent anomaly may be due to the prerequisite of relevant adult knowledge and experience to recognize and appreciate UGS’ social benefits. Physical health benefits, such as maternal and fetal health, morbidity and mortality rates, and other social benefits, such as quality of life, neighborhood relations, violence and crime reduction, social harmony, and city image are regarded as relatively high-order knowledge realms. A learning curve must be overcome to understand these somewhat esoteric, less direct, abstract ideas. Such knowledge and understanding might not be adequately and coherently acquired in childhood, whether in cities, suburbs, or villages. Our results were similar to those of Li et al., who studied students’ perception of campus green space in Hangzhou [90]. They found that the growth environment (growing up in rural areas, small towns, cities, or suburbs) had little impact on perceptions of social benefits provided by campus green space, such as safety.

The current living environments of residents affected their perceptions of UGS’ social benefits. Suburb residents had a significantly higher understanding of reducing demands for pain relief medication and bearing babies with larger heads than city residents. City residents had a significantly higher understanding of the considerable and sustained returns of investments in UGS than village residents. City residents had a significantly higher agreement of accelerating patient recovery than village residents. Unlike our results, Riechers et al. found that young suburb residents perceived recreation-related UGS benefits in Berlin, Germany. Young urban-core residents perceived UGS benefits related to social relations [52]. These differences may be related to greenspace layout. Some districts in cities and suburbs may have better formal public green spaces in residential areas and hospital grounds. The uneven and often unequal provision and access to green spaces may affect residents’ perceptions of social benefits in different living environments [91,92,93]. Urban planners could comprehensively assess UGS’s social and spatial distribution, the efficiency and fairness of green space provision, and develop alleviating measures to address spatial and social injustice [50].

### 4.3. Residents with “Poor” Self-Rated Health Status Had Better Perceptions of Other UGS Social Benefits

Residents with “poor” self-rated health status had a significantly higher understanding of improving a city’s image and making cities livable and sustainable than those with “good” or “fair” ratings. High-quality UGS can improve a city’s image, attract investments, and offer attractive and livable abodes [46,94,95]. Some modern hospitals and health centers have installed therapeutic gardens and other green spaces to meet patients’ physiological and psychological needs [96]. As an essential component of the urban ecosystem, UGS is increasingly used by residents seeking outdoor venues to improve their health. Further studies could quantify users’ preference for the treatment landscape to refine the design to meet users’ needs.

However, the residents with “excellent” self-rated health status had a significantly lower understanding of bearing babies with larger heads than those with “good” or “fair” ratings. This finding diverged from Romagosa, who found that tourists rating themselves healthy could better perceive the park’s health benefits [57]. It may be difficult to perceive the relationship between UGS visits and neonatal head development. Studies have found a causal connection between pregnant women living in a healthy green environment and birth outcomes [97]. However, the underlying mechanisms of such benefits are yet to be ascertained. Continued research of UGS’ benefits can cover different life stages from prenatal to the elderly. Translation of more findings to general consumption could raise awareness, and enhance perceptions, hopefully leading to better provision and better benefits for inclusive patronage by different social groups.

### 4.4. Policy Implications, Future Studies and Limitations

The study yields interesting findings with potential for application and hints for further studies. Some pertinent issues are elaborated below. The results indicate that respondents recognize the health benefits of UGS for themselves and their contemporaries. Such a time-bracketed perception profile does not deviate from the outcome of studies elsewhere and expectations. However, their understanding of prenatal values is deemed inadequate and fuzzy. The increase in birth weight and head circumference, and the higher probability of full-term pregnancy can bring transgenerational benefits. They have far-reaching implications on the long-term health of the babies in both physical and mental terms.

The development of these pre-strengthened children can be characterized by fewer health problems and better aptitude for scholastic achievements in their formative years. If such long-term wellbeing dividends can influence more people, the community’s overall health status and related quality of life could be correspondingly raised. Collaterally, the healthcare expenditures could be lowered, and the savings could be diverted to other pressing needs. This non-monetary measure of UGS gains to society could be assessed by research, and the findings could be more proactively promoted using the citizen science approach.

In terms of policy, expectant mothers could be coached and encouraged to visit parks regularly. This advocacy must be accompanied by the provision of more UGS that are accessible and proximal to more homes. Moreover, the quality of the greenery, which is the raison d’être of the health benefits, should be appropriately upgraded regarding landscape design, species choice and combination, and venue management. In the interest of environmental justice, the low-income neighborhoods, often beset by deprivation of the critical UGS wellbeing resources, should receive more attention. The vicious circle of economic poverty, aggravated by UGS and health poverty, could be broken. For the well-endowed precincts, the strategy should be to enhance the public education endeavors to lure more pregnant women to visit green spaces.

For Beijing, with an above-average provision of UGS area, the main concern is the uneven distribution pattern and site quality, rather than quantity. Therefore, public policy could be steered towards improving access to more people. New residential developments could be consciously located at or near places with rich natural endowments. Spatial ecological planning could be adopted to preserve high-quality natural enclaves within and adjacent to the built-up areas. Where inheritance of pre-urbanization nature is unavailable or infeasible, high-caliber UGS could be liberally created instead. To maximize benefits and accessibility, the UGS could form a spatially permeating network with a high level of connectivity that enmeshes the developed pockets. Preferably, residents should reach a green venue within about 15 min of walking.

The finding that people with self-rated poor health have a better appreciation of UGS health benefits demands some follow-up actions. This association could mean that unhealthy people are more inclined to take advantage of the salubrious effects, whereas healthy ones are not so forthcoming. Understandably, people with a suboptimal health condition or who are recuperating after illness demand some therapeutic treatments, including more exposure to a pleasant green ambiance. However, at the macro-scale consideration, people should obtain health benefits from visiting parks to prevent illness and physiological decline. It will be desirable to mold people’s attitudes and behaviors so that they are more earnest in tapping the health-facilitating, defending and prophylactic attributes of UGS. The message that health should be protected and preserved, and the realization that park visits can contribute to this crucial cause, should be more emphatically brought home.

Concerning UGS design and management, there are gaps between science and policy applications. The above concerns indicate the need for in-depth investigations by applied research projects. Besides acquiring a deeper understanding of the multiple factors leading to the differential perceptions, expectations, and behaviors, exploring practical ways to transfer and propagate knowledge are worthwhile. UGS is a universal phenomenon, and so are UGS visitors. Nevertheless, there is a lack of a high-level and comprehensive paradigm to optimize the delivery of benefits. With increasing health consciousness and climate-change impacts, the demands on UGS as a cost-effective health resource will be expected to rise continually. It is high time that the two domains and their synergistic interactions are enhanced by application of new knowledge and practices.

The scale and aims of the study must focus on our defined range of objectives. Some potentially critical issues could not be covered; such limitations could be tackled in future studies. The questionnaire survey could be expanded by splitting the sample of people living in the core city area versus the lower density suburban areas. The difference between old and new residential areas with different green space provisions could be explored. The diverse landscape designs and styles influencing people’s perceptions could be investigated. The effect of formal urban parks vis-à-vis informal urban woodlands could be evaluated. As our study was conducted in summer, the effect of the harsh winter weather on perception and health conditions could be studied.

## 5. Conclusions

This study found that residents’ perceptions of UGS physical health benefits were not high, and they were more able to perceive other social benefits. Childhood living environments did not affect perceptions of social benefits, but current living environments did. Residents with “poor” self-rated health status had higher perceptions of other social benefits, whereas residents with “excellent” ratings did not fully understand UGS physical health benefits. The findings could provide valuable information for UGS planning and development. The implications of public policies regarding UGS planning and management were discussed. The issues that demand more research and attention were identified.

First, the UGS elements related to health promotion could be more assiduously integrated into UGS’ design and publicity programs. Knowledge about health benefits could be transmitted effectively to citizens through public education and community activities in the citizen-science language. Measures could be adopted to raise patronage to bring benefits to more people. Efforts could be directed to improve greenery quality and the spatial patterns of green venues vis-à-vis population concentrations. Efforts could be directed towards creating a competitive and well-recognized green city brand, fostered by high-quality and health-promoting UGS.

Second, the social and spatial distribution of UGS could receive more attention. The design and location of UGS could ensure inclusive and equitable access by different community sectors to address a critical aspect of environmental injustice. Access to UGS situated near homes, preferably within several hundred meters, can encourage and facilitate visits and spread the benefits to more people. Third, residents’ preference for a health-enhancing and therapeutic landscape could be further investigated to guide the design of user-oriented UGS. Fourth, research of UGS’ benefits on pregnancy outcomes and at all growth stages of individuals could be deepened to bring more targeted inter-generational and life-long benefits of UGS to more people.

## Figures and Tables

**Figure 1 ijerph-19-10134-f001:**
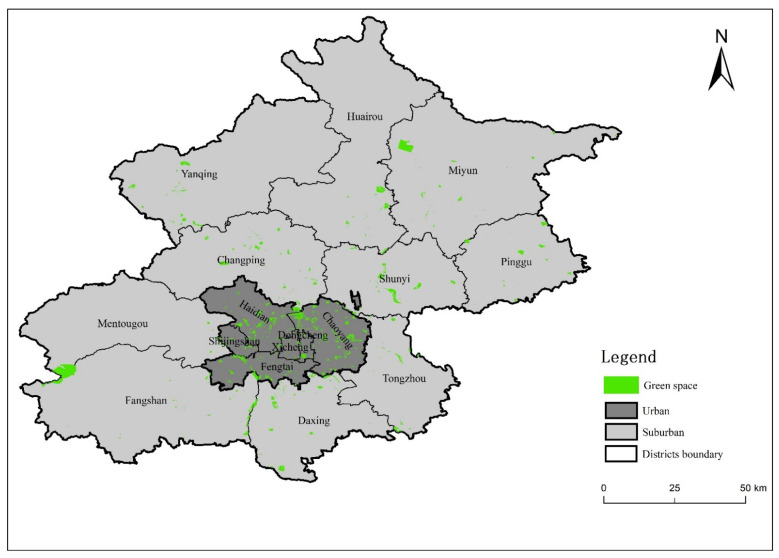
The distribution of main public urban green spaces in Beijing’s urban and suburban districts.

**Table 1 ijerph-19-10134-t001:** Classification of UGS’ social benefits into two main categories in the questionnaire survey.

B1 Physical Health Benefit		B2 Other Social Benefit
Benefits to hospital patients	B1-1 Accelerate patient recovery	B2-1 Improve quality of environment and life by UGS area
	B1-2 Reduce the need for nursing care	B2-2 Improve quality of environment and life by UGS quality
	B1-3 Reduce demands for pain relief medication	B2-3 Facilitate social networking and mutual support
	B1-4 Lower overall medical costs	B2-4 Provide calming effect to reduce outbursts of anger
Benefits to pregnancy outcomes	B1-5 Pregnant women who have more contact with nature	B2-5 Promote constructive reasoning to resolve conflicts
	give birth to babies with higher weight	B2-6 Foster social cohesion and harmony
	B1-6 Pregnant women who have more contact with nature	B2-7 Ancient trees are a symbol of urban identity and history
	give birth to babies with larger heads	B2-8 Improve a city’s image
	B1-7 Pregnant women who have more contact with nature	B2-9 Make cities livable and sustainable
	give birth to full-term babies	B2-10 Have considerable and sustained returns and are worth
Benefits to morbidity and mortality rate	B1-8 Reduce incidence rate	increasing the public expenditure
	B1-9 Reduce mortality rate	
	B1-10 Low-income residents’ more contact with nature will reduce mortality [73]	

**Table 2 ijerph-19-10134-t002:** Differences in the main socioeconomic characteristics between the survey respondents and Beijing residents based on the Chi-square test.

Socioeconomic Characteristics	χ^2^	*p*-Value
Gender	0.004	0.950
Age	72.661	0.001 **
Marriage	4.620	0.099
Education	352.000	0.001 **

The items with a significant level of <0.05 were listed. ** means *p* < 0.01.

**Table 3 ijerph-19-10134-t003:** The understanding and agreement degrees of respondents concerning UGS’ social benefits.

				Understanding Degree ^a^ (%)							Agreement Degree ^b^ (%)									
Benefit ^c^	5	4	3	2	1	Mean	SD	MD ^e^	RI ^e^	Rank ^d^	5	4	3	2	1	Mean	SD	MD ^e^	RI ^e^	Rank ^d^
**B1-1**	**13.19**	**41.90**	**31.71**	**12.04**	**1.16**	**3.54**	**0.91**	**4.00**	**0.71**	**1**	**26.85**	**44.68**	**22.92**	**5.09**	**0.46**	**3.92**	**0.86**	**4.00**	**0.79**	**1**
B1-2	6.25	22.45	41.44	25.23	4.63	3.00	0.96	3.00	0.60	4	14.35	23.61	41.20	17.13	3.70	3.28	1.03	3.00	0.66	5
B1-3	6.02	22.22	38.89	27.55	5.32	2.96	0.98	3.00	0.59	5	14.58	25.69	35.65	20.37	3.70	3.27	1.06	3.00	0.65	6
B1-4	4.40	19.44	31.71	37.50	6.94	2.77	0.99	3.00	0.55	8	11.81	24.31	36.34	21.76	5.79	3.15	1.07	3.00	0.63	9
**B1-5**	**5.79**	**16.67**	**27.08**	**37.04**	**13.43**	**2.64**	**1.09**	**2.00**	**0.53**	**10**	**10.42**	**22.45**	**39.35**	**21.06**	**6.71**	**3.09**	**1.06**	3.00	**0.62**	**10**
**B1-6**	**3.47**	**10.42**	**26.39**	**42.36**	**17.36**	**2.40**	**1.00**	**2.00**	**0.48**	**11**	**8.33**	**18.06**	**39.35**	**27.55**	**6.71**	**2.94**	**1.03**	3.00	**0.59**	**11**
B1-7	6.25	17.36	27.78	34.72	13.89	2.67	1.17	3.00	0.54	9	13.19	25.93	37.27	18.75	4.86	3.24	1.06	3.00	0.65	8
**B1-8**	**11.57**	**25.46**	**38.66**	**20.14**	**4.17**	**3.20**	**1.02**	**3.00**	**0.64**	**2**	**22.92**	**34.03**	**32.41**	**9.26**	**1.39**	**3.68**	**0.97**	**4.00**	**0.74**	**2**
**B1-9**	**8.80**	**22.22**	**38.66**	**25.46**	**4.86**	**3.05**	**1.01**	**3.00**	**0.61**	**3**	**18.06**	**31.02**	**35.19**	**14.12**	**1.62**	**3.50**	**1.00**	**3.00**	**0.70**	**3**
B1-10	6.48	19.68	31.94	32.41	9.49	2.81	1.06	3.00	0.56	7	15.28	25.46	32.87	20.83	5.56	3.24	1.11	3.00	0.65	7
Mean	7.22	21.78	33.43	29.44	8.13	2.91	1.01	2.90	0.58	6	15.58	27.52	35.25	17.59	4.05	3.33	1.03	3.20	0.67	4
**B2-1**	**31.94**	**46.06**	**17.59**	**4.40**	**0.00**	**4.06**	**0.82**	**4.00**	**0.81**	**1**	**43.29**	**43.98**	**10.88**	**1.85**	**0.00**	**4.29**	**0.73**	**4.00**	**0.86**	**2**
**B2-2**	**33.33**	**42.82**	**19.68**	**4.17**	**0.00**	**4.05**	**0.83**	**4.00**	**0.81**	**2**	**45.14**	**41.44**	**11.34**	**1.85**	**0.23**	**4.29**	**0.76**	**4.00**	**0.86**	**1**
B2-3	15.28	39.12	36.11	9.03	0.46	3.60	0.87	4.00	0.72	8	25.69	41.44	26.39	5.56	0.93	3.85	0.90	**4.00**	0.77	9
**B2-4**	**15.51**	**35.65**	**32.64**	**14.35**	**1.85**	**3.49**	**0.98**	**4.00**	**0.70**	**10**	**28.70**	**33.10**	**28.70**	**7.87**	**1.62**	**3.79**	**1.00**	**4.00**	**0.76**	**10**
**B2-5**	**10.42**	**29.40**	**35.88**	**21.53**	**2.78**	**3.23**	**0.99**	**3.00**	**0.65**	**11**	**19.44**	**31.48**	**34.03**	**12.96**	**2.08**	**3.53**	**1.02**	**4.00**	**0.71**	**11**
B2-6	15.97	38.43	32.64	11.81	1.16	3.56	0.93	4.00	0.71	9	28.01	38.89	25.46	6.25	1.39	3.86	0.94	**4.00**	0.77	8
B2-7	28.24	42.13	22.22	6.94	0.46	3.91	0.90	4.00	0.78	5	43.75	37.04	15.05	3.24	0.93	4.19	0.88	**4.00**	0.84	5
B2-8	31.71	43.29	20.14	4.63	0.23	4.02	0.85	4.00	0.80	4	45.14	39.12	11.34	3.70	0.69	4.24	0.85	**4.00**	0.85	3
B2-9	31.25	45.37	18.29	5.09	0.00	4.03	0.84	4.00	0.81	3	42.59	39.12	15.05	3.01	0.23	4.21	0.82	**4.00**	0.84	4
B2-10	21.53	40.28	25.93	11.11	1.16	3.70	0.97	4.00	0.74	7	34.03	37.27	21.53	5.79	1.39	3.97	0.96	**4.00**	0.79	7
Mean	23.52	40.25	26.11	9.31	0.81	3.76	0.90	3.90	0.75	6	35.58	38.29	19.98	5.21	0.95	4.02	0.89	4.00	0.80	6

^a^ Understanding degree was coded on an ordinal scale: 5-excellent; 4-good; 3-fair; 2-limited; 1-don’t know. ^b^ Agreement degree was coded on an ordinal scale: 5-very strong; 4-strong; 3-moderate; 2-weak; 1-don’t agree. ^c^ Values in bold font indicate a higher or lower degree of understanding or agreement. ^d^ The UGS’ social benefits within each category (B1 and B2) were ranked by the mean value. ^e^ SD refers to standard deviation, MD to median, RI to relative importance.

**Table 4 ijerph-19-10134-t004:** The impacts of childhood living environments on respondents’ perceptions of UGS’ social benefits.

Social Benefit	Understanding Degree ^a^	Agreement Degree ^a^
B1-1	Chi-square = 1.947, *p* = 0.378	Chi-square = 0.600, *p* = 0.741
B1-2	Chi-square = 0.554, *p* = 0.758	Chi-square = 0.552, *p* = 0.759
B1-3	Chi-square = 0.369, *p* = 0.832	Chi-square = 0.475, *p* = 0.789
B1-4	Chi-square = 0.125, *p* = 0.940	Chi-square = 0.531, *p* = 0.767
B1-5	Chi-square = 0.736, *p* = 0.692	Chi-square = 1.828, *p* = 0.401
B1-6	Chi-square = 1.259, *p* = 0.533	Chi-square = 0.522, *p* = 0.770
B1-7	Chi-square = 4.425, *p* = 0.109	Chi-square = 0.334, *p* = 0.846
B1-8	Chi-square = 0.228, *p* = 0.892	Chi-square = 0.435, *p* = 0.805
B1-9	Chi-square = 0.862, *p* = 0.650	Chi-square = 0.575, *p* = 0.750
B1-10	Chi-square = 0.145, *p* = 0.930	Chi-square = 0.033, *p* = 0.984
B2-1	Chi-square = 0.957, *p* = 0.620	Chi-square = 0.698, *p* = 0.705
B2-2	Chi-square = 0.305, *p* = 0.859	Chi-square = 0.050, *p* = 0.975
B2-3	Chi-square = 4.038, *p* = 0.133	Chi-square = 1.445, *p* = 0.486
B2-4	Chi-square = 3.841, *p* = 0.147	Chi-square = 3.955, *p* = 0.138
B2-5	Chi-square = 2.135, *p* = 0.344	Chi-square = 5.798, *p* = 0.055
B2-6	Chi-square = 3.226, *p* = 0.199	Chi-square = 0.452, *p* = 0.798
B2-7	Chi-square = 1.949, *p* = 0.377	Chi-square = 1.344, *p* = 0.511
B2-8	Chi-square = 0.805, *p* = 0.669	Chi-square = 0.422, *p* = 0.810
B2-9	Chi-square = 1.078, *p* = 0.583	Chi-square = 0.458, *p* = 0.795
B2-10	Chi-square = 1.575, *p* = 0.455	Chi-square = 1.739, *p* = 0.419

^a^ Non-parametric statistical tests were employed to assess the differences between social benefits. The Kruskal–Wallis test was applied first. If the results showed significant differences, the Mann–Whitney test was then applied for intra-group comparison.

**Table 5 ijerph-19-10134-t005:** The impacts of current living environments on respondents’ perceptions of UGS’ social benefits.

Social Benefit	Understanding Degree ^a^	Agreement Degree ^a^
B1-1	Chi-square = 2.600, *p* = 0.273	Chi-square = 6.741, *p* = 0.034 * **n1** × n3-(U = 5612.500, *p* = 0.025 *) **n2** × n3-(U = 558.500, *p* = 0.013 *)
B1-2	Chi-square = 5.069, *p* = 0.079	Chi-square = 0.346, *p* = 0.841
B1-3	Chi-square = 8.079, *p* = 0.018 * N1 × **N2**-(U = 5236.000, *p* = 0.005 **)	Chi-square = 0.151, *p* = 0.927
B1-4	Chi-square = 4.427, *p* = 0.109	Chi-square = 0.755, *p* = 0.686
B1-5	Chi-square = 3.456, *p* = 0.178	Chi-square = 0.302, *p* = 0.860
B1-6	Chi-square = 8.287, *p* = 0.016 * N1 × **N2**-(U = 5455.000, *p* = 0.014 *)	Chi-square = 1.473, *p* = 0.479
B1-7	Chi-square = 3.546, *p* = 0.170	Chi-square = 1.546, *p* = 0.462
B1-8	Chi-square = 0.475, *p* = 0.789	Chi-square = 0.707, *p* = 0.702
B1-9	Chi-square = 0.122, *p* = 0.941	Chi-square = 0.114, *p* = 0.944
B1-10	Chi-square = 5.274, *p* = 0.072	Chi-square = 0.469, *p* = 0.791
B2-1	Chi-square = 4.363, *p* = 0.113	Chi-square = 0.254, *p* = 0.881
B2-2	Chi-square = 5.517, *p* = 0.063	Chi-square = 1.028, *p* = 0.598
B2-3	Chi-square = 0.132, *p* = 0.936	Chi-square = 0.254, *p* = 0.881
B2-4	Chi-square = 0.969, *p* = 0.616	Chi-square = 3.009, *p* = 0.222
B2-5	Chi-square = 1.599, *p* = 0.450	Chi-square = 0.489, *p* = 0.783
B2-6	Chi-square = 1.768, *p* = 0.413	Chi-square = 3.434, *p* = 0.180
B2-7	Chi-square = 0.918, *p* = 0.632	Chi-square = 4.893, *p* = 0.087
B2-8	Chi-square = 1.497, *p* = 0.473	Chi-square = 5.727, *p* = 0.057
B2-9	Chi-square = 3.182, *p* = 0.204	Chi-square = 2.132, *p* = 0.344
B2-10	Chi-square = 6.441, *p* = 0.040 * **N1** × N3-(U = 5437.000, *p* = 0.013 *)	Chi-square = 4.077, *p* = 0.130

^a^ Non-parametric statistical tests were employed to assess the differences between social benefits. The Kruskal–Wallis test was applied first. If the results showed significant differences, the Mann–Whitney test was then applied for intra-group comparison. The items with a significant level of <0.05 were listed. * means *p* < 0.05, ** means *p* < 0.01, and boldface indicates a higher average term. N1, N2, and N3 represent the understanding levels of respondents currently living in the city, suburb, and village, respectively; n1, n2, and n3 represent the agreement levels of respondents currently living in the city, suburb, and village, respectively.

**Table 6 ijerph-19-10134-t006:** The impacts of self-rated health status on respondents’ perceptions of UGS’ social benefits.

Social Benefit	Understanding Degree ^a^	Agreement Degree ^a^
B1-1	Chi-square = 1.581, *p* = 0.664	Chi-square = 3.020, *p* = 0.389
B1-2	Chi-square = 5.926, *p* = 0.115	Chi-square = 1.495, *p* = 0.684
B1-3	Chi-square = 2.968, *p* = 0.397	Chi-square = 5.518, *p* = 0.138
B1-4	Chi-square = 0.900, *p* = 0.825	Chi-square = 6.817, *p* = 0.078
B1-5	Chi-square = 3.895, *p* = 0.273	Chi-square = 4.488, *p* = 0.213
B1-6	Chi-square = 8.406, *p* = 0.038 * H1 × **H2**-(U = 9206.000, *p* = 0.033 *) H1 × **H3**-(U = 5023.500, *p* = 0.010 *)	Chi-square = 7.148, *p* = 0.067
B1-7	Chi-square = 4.683, *p* = 0.197	Chi-square = 2.832, *p* = 0.418
B1-8	Chi-square = 4.638, *p* = 0.200	Chi-square = 2.450, *p* = 0.484
B1-9	Chi-square = 1.197, *p* = 0.754	Chi-square = 0.606, *p* = 0.895
B1-10	Chi-square = 6.341, *p* = 0.096	Chi-square = 2.815, *p* = 0.421
B2-1	Chi-square = 6.235, *p* = 0.101	Chi-square = 3.867, *p* = 0.276
B2-2	Chi-square = 6.354, *p* = 0.096	Chi-square = 5.077, *p* = 0.166
B2-3	Chi-square = 2.954, *p* = 0.399	Chi-square = 3.089, *p* = 0.378
B2-4	Chi-square = 6.785, *p* = 0.079	Chi-square = 1.123, *p* = 0.772
B2-5	Chi-square = 5.689, *p* = 0.128	Chi-square = 0.482, *p* = 0.923
B2-6	Chi-square = 4.897, *p* = 0.179	Chi-square = 1.800, *p* = 0.615
B2-7	Chi-square = 6.756, *p* = 0.080	Chi-square = 2.466, *p* = 0.481
B2-8	Chi-square = 12.109, *p* = 0.007 ** **H1** × H2-(U = 9009.000, *p* = 0.015 *) **H1** × H3-(U = 5134.500, *p* = 0.018 *) H2 × **H4**-(U = 219.000, *p* = 0.020 *) H3 × **H4**-(U = 120.500, *p* = 0.017 *)	Chi-square = 8.896, *p* = 0.031 * **h1** × h2-(U = 8740.000, *p* = 0.004 **) **h1** × h3-(U = 5229.000, *p* = 0.027 *)
B2-9	Chi-square = 13.438, *p* = 0.004 ** **H1** × H2-(U = 4942.500, *p* = 0.005 **) H2 × **H4**-(U = 222.500, *p* = 0.021 *) H3 × **H4**-(U = 114.500, *p* = 0.013 *)	Chi-square = 6.093, *p* = 0.107
B2-10	Chi-square = 3.057, *p* = 0.383	Chi-square = 1.609, *p* = 0.657

^a^ Non-parametric statistical tests were employed to assess the differences between social benefits. The Kruskal–Wallis test was applied first. If the results showed significant differences, the Mann–Whitney test was then applied for intra-group comparison. The items with a significant level of <0.05 were listed. * means *p* < 0.05, ** means *p* < 0.01, and boldface indicates a higher average term. H1, H2, H3, and H4 represent the understanding level of respondents with excellent, good, fair, and bad self-rated health status, respectively; h1, h2, and h3 represent the agreement level of respondents with excellent, good, and fair self-rated health status, respectively.

## Data Availability

Data available upon request sent to the corresponding author.

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
