# Peer review of "Effects of Self-Rated Health Status on Residents’ Social-Benefit Perceptions of Urban Green Space"

_ijerph, 2022, doi:10.3390/ijerph191610134_

Round 1

Reviewer 1 Report

Dear authors

The article fits the theme of the journal, but seems to be outdated due to the survey conducted in 2018. I would have a few comments on this:
- It contains an acceptable literature review
- A map of urban green spaces (UGS) in Beijing should be more meaningful than a mere description in the text,
- The text indicates that the percentage of green space is very high (80%), but it is not clear how it was determined. Please provide a graphical representation to aid understanding.
- Are the 500 questionnaires sufficient for the more than 21 million inhabitants?
- This article provides an analysis of a questionnaire survey conducted in the summer of 2018.
- Table 1 shows the classification of UGS social services into two main categories of the questionnaire survey: B1 and B2.
- However, Figure 1 shows the average understanding and agreement scores of B1, B2, B3, and B4, although B3 and B4 were not described in the text or in Table 1.
-In the conclusion it states "More attention could be paid to the spatial distribution of UGS", but it is not clear how this survey could improve the design of UGS, please specify.

Reviewer 2 Report

This study investigated residents' perception of the benefit of UGS in Beijing based on a questionnaire survey. Although the topic has a certain level of importance, this work has provided few innovations. The methodology is simple and some contents are very hard to follow due to the improper English expression. Polishing the English is a must before the consideration of publication.

There are some interesting findings in this work: people recognize the psychological benefit of UGS rather than its physical health benefit. However, as the questionnaire was conducted on only 439 people, which is a very small group compared with the population in Beijing, these findings are highly questionable. Also, some perceptions seem hard to be verified in this study. For example, whether UGS can help reduce the mortality rate should be further verified by statistical data. I don't think any ordinary citizen can remember the number when he/she did this questionnaire. 

In summary, the current work is not suitable for publication in high-level academic journals. The findings of this work should be verified by introducing external reference data, or increasing the number of respondents significantly to enhance their reliability, even I don't take this as a proper way to conduct this study.  

Reviewer 3 Report

1. The authors administered questionnaires to 432 Beijing residents only. It would be interesting to consider other cities outside Beijing in China. The sample size is reasonable. However, more correspondents could be included in the generalization of the projected results.

2. The percentage of Beijing residents in the income category obtained in 2017 was used in this study. Currently, there is a significant difference between the values in 2017 and today (2022). Any reason for using the data of 2017? It would be more realistic to use recent data for analysis and inference.

3. The authors need to compare the results of the current study with related results in the open literature, especially for related urban green spaces.

4. The future scope is missing in the conclusion section. It would be nice to include the future scope and research direction.

Reviewer 4 Report

The paper examined residents' perceptions of UGS social benefits and their driving factors. Before considering for publication, the paper should be improved by considering the following:

1. The current topic is confusing. The adoption of the word "can" makes the topic looks like just a statement. Authors are encouraged to rephrase the topic in line with the content of the paper.

2. There is a need to explain in details the key issues which are space, green space and urban green space. The introduction section should be structured to discuss the basic items in appropriate order.

3. There are lots of syntax and technical errors in the current paper. A proofreader should be engaged to further improve the paper. Check for instance "For the current situation of perception, the percentage of respondents in each perception level in the total number of respondents...."

4. Subject to this statement "The percentage of Beijing residents in the income categories (2017) was obtained from the website (http://www.sohu.com/a/168513231_99914104)", is this an official government website? Why adopting 2017 data for a paper to be published probably in 2022? Are there no updated information and reliable source for the population?

5. What is the basis for the collection of 500 questionnaires? What is the total population and calculated sample size?

6. There is a need to explain what makes some of the questionnaires invalid. 68 invalid questionnaires is a lot and require explanation. 

7. There is no proof of this statement "Their socio-demographic profile was similar to the residents' characteristics extracted from the Beijing Statistical Yearbook 2019 and the China Statistical Yearbook 2019" in the current version of the paper.

8. For the data analysis, there is a need to state the method adopted and rationale for adopting the same. Previous publications where the method was adopted should also be cited to justify the choice

9. Subject to this statement "Kruskal-Wallis test and Mann-Whitney test were used to further analyze residents' perceptions differences of different benefits in different living environments and self-rated health conditions", The authors need to understand the usage of MW and KW tests. The first is for 2 categories while the later is for 3 or more groups. There is also the need to explain the groups and explain the rationale for comparing their results. 

10. For the results, since a Likert scale was adopted, there is a need to adopt such descriptive statistics as MIS, SD and RI to determine the average score of each variables and not just percentage. 

11. For the analysis, tables are more appropriate as it will depict more information.

12. The discussion section should be improved by explaining the implications of the study

13. For the conclusion, the section should further explain limitations of the study, areas for further study as well as recommendations. The recommendations should be directed at stakeholders that will find the results useful and how they can apply the same.

14. The references should be updated to include publications in 2022.

Round 2

Reviewer 1 Report

Dear authors

I understand your justification, but I advise you not to publish the results of a study so late after the survey in the future.

Also, I think that the analysis of accessibility through the buffer zone is not enough, because we have access only through roads or alleys, so I advise not to use this method in the future.

Reviewer 2 Report

I did not see much improvement in this revised version. The study lacks a solid experimental design so that the results are not very convincing. Besides the concerns about the small sample size, the authors did not give any response to my question about the reliability of the contents in their questionnaire (my comments: For example, whether UGS can help reduce the mortality rate should be further verified by statistical data. I don't think any ordinary citizen can remember the number when he/she did this questionnaire.)
